# RetinaGS: Scalable Training for Dense Scene Rendering with Billion-Scale 3D Gaussians

## Abstract

In this work, we explore the possibility of training high-parameter 3D Gaussian splatting (3DGS) models on large-scale, high-resolution datasets. We design a general model parallel training method for 3DGS, named RetinaGS, which uses a proper rendering equation and can be applied to any scene and arbitrary distribution of Gaussian primitives. It enables us to explore the scaling behavior of 3DGS in terms of primitive numbers and training resolutions that were difficult to explore before and surpass previous state-of-the-art reconstruction quality. We observe a clear positive trend of increasing visual quality when increasing primitive numbers with our method. We also demonstrate the first attempt at training a 3DGS model with more than one billion primitives on the full MatrixCity dataset that attains a promising visual quality.

## 1 Introduction

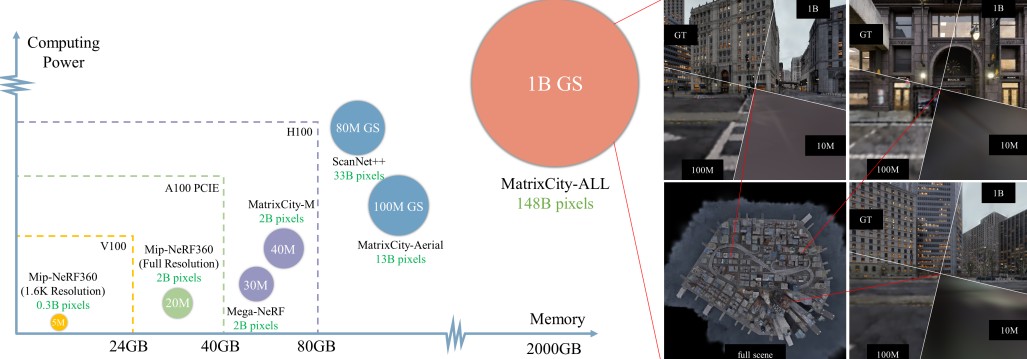

Figure 1: **Left:** Different sizes of datasets require varying levels of computational power and numbers of 3DGS Primitives. Larger and higher-resolution datasets can no longer be trained using just a single GPU, which limits the pursuit of scale and fidelity in 3DGS reconstruction. **Right:** The billion-level model bring better visual experience than million-level model on MatrixCity-ALL dataset, which is trained via our distributed modeling with 64 GPUs.

3D scene reconstruction with Gaussian Splatting (GS)(Kerbl et al., 2023) has drastically improved rendering quality and rendering speed over previous neural 3D representation(Mildenhall et al., 2021; Zhang et al., 2020; Chen et al., 2022). However, this success has been largely limited to reconstructing scenes with limited image or video resolution (typically $<= 1600$ pixels wide), data volume, and view distance. Viewing the scene at high resolution or close range remains an unsolved challenge. To achieve imaging effects flawless to the human retina, which we refer to as the goal of *retina-level reconstruction*, we would desire to train the GS models with higher spatial resolution, larger datasets, and more varying viewing distances, as illustrated in Fig. 1 (right).

Despite the success in scaling up machine learning models (Vaswani et al., 2017; Radford et al., 2019; Brown et al., 2020). Training of 3DGS based reconstruction models have largely been limited to a single GPU. However, in Fig. 1 we can see that the time and memory footprint needed for 3DGS training on moderately size scene quickly grows infeasible for even the best single GPUs. To

overcome the limitation of single GPUs, recent works (Lin et al., 2024; Liu et al., 2024) resort to approximately distributed training for specific data distribution, i.e., the bird's-eye view city data. These methods train independent 3DGS models in parallel on multiple GPUs, where each submodel is responsible for one subspace of the dataset. These subspaces are carefully partitioned to minimize the chance that a ray passes through multiple subspaces, and thus to ensure that the individual models' rendering output can be concatenated to represent a large-scale scene. For bird's-eye view datasets, planar cells exist as a convenient choice of the partition. However, when the subspace partition for "one subspace per ray" is infeasible, as shown in Fig. 2, these approximate distributed training methods will either meet difficulty in training or lead to visible artifacts in rendering.

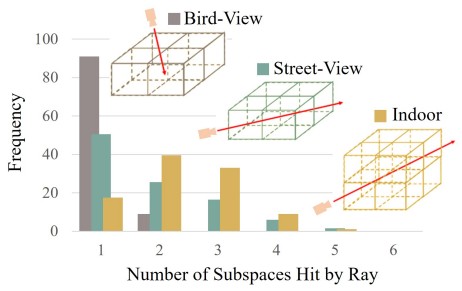 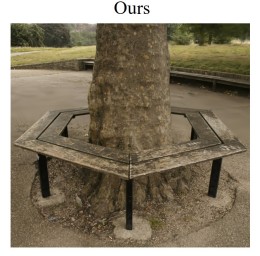 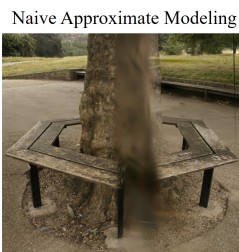

Figure 2: **Left:** Histogram about number of subspaces hit by each ray. We analyzed three datasets: the Matrixcity(Li et al., 2023b) bird-view dataset, the Matrixcity street-view dataset, and the Scan-net++(Yeshwanth et al., 2023) indoor dataset. All scenes were divided into 8 subspaces along the x, y, and z directions. The statistics reveal that these datasets exhibit very different data distributions. **Right:** Visual comparison of our distributed model and previous approximate distributed models.

In this paper, we present a method for distributed training of GS models that retains the exact equivalence to the existing single-GPU training scheme, while not relying on a certain data distribution. We start with the fact that the model space can be divided into a set of non-overlaping convex subspaces and we can identify one subset of splats for each subspace so that all the subsets collectively form an overlapping cover of the set of all splats. We then show that with a proper subset assignment strategy, each ray's original iterative alpha-blending process can be formulated in a hierarchical manner. In this formulation, which applies to **any** distribution of splats and model space, we can first compute partial color and alpha values for each subset and then obtain the rendered pixel color value by orderly merging these partial values.

We derive our training method based on this formulation and designate one worker for each subset. In the forward process, all workers compute subset-level color and alpha values for all rays in parallel. The partial values, sufficient to compute rendered pixel values, can then be merged through cross-worker communication with a minimal message size. After computing the reconstruction loss and its gradients w.r.t. the partial values, we can distribute the corresponding gradients to each worker and run the backward process in parallel. We further improve training efficiency by utilizing KD-tree to produce subspaces that induce subsets with more even cardinality.

We validate the training method on multiple 3D reconstruction datasets with high resolutions training. The trained GS models outperform single GPU trained 3DGS (Kerbl et al., 2023) and several baseline distirbuted 3DGS training approaches on Mip-NeRF360(Barron et al., 2022), Mega-NeRF (Turki et al., 2022), ScanNet++(Yeshwanth et al., 2023) and MatrixCity(Li et al., 2023b) datasets. Through scaling study, we observe a clear tendency to improve visual quality when the GS models grow in splat numbers. Finally, we demonstrate to our knowledge the first attempt at training GS models with billion-scale primitives on city-level dataset with 140k images, resulting in unprecedented visual experience using million-scale primitives.

## 2 RELATED WORK

**Distributed Neural Radiance Fields.** NeRF (Mildenhall et al., 2021) has revolutionized 3D scene reconstruction and novel viewpoint generation with its photorealistic rendering capabilities, sparking a series of subsequent innovations focused on enhancing various aspects of the technology. These

advancements strive to improve rendering quality (Zhang et al., 2020; Baumgartner et al., 2004; Barron et al., 2021; 2022), optimize training speed (Sun et al., 2022; Fridovich-Keil et al., 2022; Chen et al., 2022; Müller et al., 2022), increase memory efficiency (Reiser et al., 2023; Li et al., 2023a; Rho et al., 2023) and expand the scale of reconstructed scenes (Zhenxing & Xu, 2022; Xiangli et al., 2022; Tancik et al., 2022; Turki et al., 2022; Xu et al., 2023; Zhang et al., 2023; Song & Zhang, 2023; Wu et al., 2023). While these fields might benefit from larger models or enhanced computational power, it is primarily the pursuit of scaling scene sizes that has driven researchers to explore ways to speed up or scale up NeRF models. Switch-NeRF (Zhenxing & Xu, 2022) utilizes a Mixture of Experts (MoE) to increase the capacity of NeRF models, enabling them to represent urban-scale scenes effectively. Bungee-NeRF (Xiangli et al., 2022) employs a hierarchical assembly of submodels, granting the model extensive multiscale representational capabilities. Recognizing the limitations imposed by GPU computational power and memory on model scale and training velocity, some researchers have begun deconstructing models into smaller components, adopting distributed approaches to address these challenges. Block-NeRF (Tancik et al., 2022) segments cities into multiple overlapping blocks, each represented by its own NeRF model, and uses neural networks to fuse the outputs of multiple NeRF models in image space, achieving seamless visual results. Mega-NeRF (Turki et al., 2022) introduces a simple geometric clustering algorithm and partitions training pixels into various NeRF submodules, which can then be trained in parallel. All the distributed methods mentioned above only perform 2D planar cell partitioning on the ground, which is suitable for flat urban scenes. However, for detailed rendering of typical indoor scenes and scenes that have a combination for multiple view types, large model capacity is required in both the vertical and horizontal directions, making these methods insufficient for the task.

**Distributed Point-Based Representation.**    NeRF utilizes a volumetric rendering approach that inherently limits their inference speed, which makes it challenging to achieve real-time performance. Several studies (Yu et al., 2021; Reiser et al., 2023; Yariv et al., 2023; Tang et al., 2023) have focused on optimizing inference speeds, yet achieving both real-time performance and high-quality rendering remains elusive. In contrast, the 3D Gaussian Splatting (3DGS) (Kerbl et al., 2023) method, which employs a point-based scene representation, achieves state-of-the-art rendering quality and speed and has significant advantages in training speed. However, the "shallow and wide" structure of the 3DGS model and its explicit representation lead to a larger parameter count and increased training memory footprint. VastGaussian (Lin et al., 2024) is the first attempt to large-scale scene reconstruction with 3DGS. It divides the ground plane into a series of 2D cells with one splat subset for each cell. Unlike our method, all subsets are mutually exclusive and considered as independent 3DGS models in training. To mitigate the deviation from the original 3DGS rendering equation at subspace boundaries, VastGaussian introduces additional nearby cameras and Gaussian primitives to assist in training at the cost of increased compute and memory. CityGaussian (Liu et al., 2024) uses a similar partitioning strategy but trains an additional coarse GS model as the baseline to regularize submodels. However, artifacts caused by deviation from the proper rendering equation could still intensify at certain view angles and overcome the mitigations. As a result, these approaches mostly assume bird-eye-view urban scenarios due to their simplicity in view angles and splat distribution. Our method is based on an equivalent form of the 3DGS rendering equation. Thus it can be applied to any scene with arbitrary splat distribution.

**Distributed Deep Learning.**    Our work is also related to distributed deep learning, which aims to scale up the training system for deep neural network models (Dean et al., 2012; Abadi et al., 2015; Li et al., 2020). Early approaches revolve around training multi-GPU convolutional neural networks (CNNs) (Krizhevsky et al., 2012) on image (Jia et al., 2014) and video (Wang et al., 2015) datasets, with data parallelism being highly efficient thanks to CNNs' high compute to bandwidth ratio. Recent development of large language models presented new challenges in distributing model parameters and larger clusters, resulting in dedicated model parallelism (Krizhevsky, 2014) approaches for Transformers (Vaswani et al., 2017) models, such as tensor-parallelism (Shoeybi et al., 2019), pipeline parallelism (Huang et al., 2018), or hybrid parallelism (Shoeybi et al., 2019; Liu & Abbeel, 2023; Lepikhin et al., 2020). Fully sharded data parallelism (FairScale authors, 2021), a type of redundancy-free data parallelism (Rajbhandari et al., 2020), also works great with LLM training. Our method is best described as model parallelism as it partitions GS model parameters onto multiple workers to distribute workload and incurs communication cost only for Gaussian on the subspace boundaries. This is well suited to GS models' relatively low compute-to-parameter ratio[1], which makes data parallelism less efficient.

---

[1]also known as operational intensity (Williams et al., 2009).

## 3 APPROACH

### 3.1 3D GAUSSIAN SPLATTING

3D Gaussian Splatting (3DGS) utilizes a series of anisotropic 3D Gaussian primitives to explicitly characterize scenes. Each Gaussian primitive, known as a *splat*, is defined by its central position $\mu \in \mathbb{R}^3$, and a covariance matrix $\Sigma \in \mathbb{R}^{3\times3}$. A splat's influence at any given point $x$ within the scene's world coordinate system is attenuated by the Gaussian function

$$G(x) = e^{-\frac{1}{2}(x-\mu)^T \Sigma^{-1}(x-\mu)}. \tag{1}$$

In practice, the function is truncated to save computation. Each splat also carries an opacity $\alpha \in \mathbb{R}$. Its color attributes $F \in \mathbb{R}^C$ are expressed through spherical harmonics (SH) $c \in \mathbb{R}^3$ to allow view-dependent textures. A view is rendered through rasterization the Gaussian primitives onto a 2D imaging plane, during which the 3D Gaussians are projected to 2D Gaussians $G'(x')$ through Jacobian linearization as described in (Zwicker et al., 2001). As a result, the influence weight of each 3D primitive on a given ray $l$ needs to be computed through path integration, whereas a 2D primitive only requires a single sampling, that is, $g(l) = G'(x')$, where $x'$ is the intersection point of the ray $l$ with the 2D imaging plane. By employing alpha-blending, these primitives are rendered with the following rendering equation

$$C(l) = \sum_{g_i \in N_l} c_i \sigma_i \prod_{g_j \in N_l,\, j<i} (1 - \sigma_j), \quad \sigma_i = \alpha_i g_i(l), \tag{2}$$

where $N_l$ denotes the set of Gaussian primitives that contribute to ray $l$, arranged in order of their depth.

### 3.2 DISTRIBUTED TRAINING OF 3DGS

Given the set of all splats $N = \{g_i\}$ and $K$ workers $\{w_1, \ldots, w_K\}$, we aim to devise a distributed training method with minimal communication overhead that still conforms to the rendering equation in Eq. 2. We first divide the scene space into a set of **convex subspaces** $S_1, \ldots, S_K$. One worker is expected to only work on a subspace and incur minimal communication needs. To achieve this, we generate $K$ subsets $N_1, \ldots, N_K$ and $N_1 \cup N_2 \cup \ldots \cup N_K = N$, which are allowed to overlap. Below we show how to create $N_k$ and manipulate Eq. 2 for distributed computation for the workers.

Given a ray $l$ and a subspace $S_k$, we can always obtain a subset of $N$ as $N_{lk}$ which denotes all splats that, when projected to a truncated 2D Gaussian according to $l$, intersects with $l$ within $S_k$. Note since the Gaussians are truncated, not all splats will intersect with $l$. Since $N_{lk}$ is ray dependent, we define $N_k^* = \cup_l N_{lk}$ as the union of $N_{lk}$ for all possible rays. Going through every possible ray to obtain $N_k^*$ is infeasible. However, we know that any intersection point must reside on its corresponding 3D ellipse. So, we can instead define $N_k$ as the set of splats whose corresponding 3D ellipsoid intersects with $S_k$. Then it is obvious that $N_k^* \subseteq N_k$ and $N_k$ is not dependent on any specific ray. To render the color for a given ray, We can first calculate the partial color $C_k(l)$ and the partial opacity $T_k(l)$ on $N_k$ as

$$C_k(l) = \sum_{g_i \in N_k} c_i \sigma_i \mathbb{1}(g_i \in N_{lk}) \prod_{g_j \in N_k,\, j<i} (1 - \sigma_j \mathbb{1}(g_j \in N_{lk})),$$

$$= \sum_{g_i \in N_{lk}} c_i \sigma_i \prod_{g_j \in N_{lk},\, j<i} (1 - \sigma_j) \tag{3}$$

$$T_k(l) = \prod_{g_i \in N_k} (1 - \sigma_i \mathbb{1}(j \in N_{lk}))$$

$$= \prod_{g_i \in N_{lk}} (1 - \sigma_i), \tag{4}$$

where $\sigma_i = \alpha_i g_i^k(l)$ and $\mathbb{1}(\cdot)$ denotes the indicator function which equals to 1 when the condition specified is true and otherwise equals to 0. Note splats in $N_k$ are ordered by their distance to the origin of $l$ to attain the index $i$, $j$ in Eq. 3 and Eq. 4. The introduction of the indicator function allows the accumulation to be carried on the entire $N_k$ for any ray $l$. Due to the convex assumption of the

subspaces, when a ray traverses a subspace, the segment of the ray within that subspace is necessarily continuous (Rockafellar, 2015). This enables us to achieve a fully equivalent and complete rendering result by simply performing a weighted sum of the computation results from each subspace along the direction of the ray's path. The partial values can then be merged in the order that the ray passes through them, which can be represented as a permutation $o_l$ of size $K$:

$$C(l) = \sum_{k \in o_l} C_k(l) \prod_{m \in o_l, \ m < k} T_m(l). \tag{5}$$

Substituting Eq. 3 and 4 into Eq. 5, we can observe that Eq. 2 is equivalent to Eq. 5 for any $l$. The relevant proof process has been placed in the appendix. Therefore we have transformed the rendering equation of 3DGS into the independent computation of subset-level partial colors and opacities and the subsequent merging. All workers can compute their corresponding partial values in parallel for each subset $N_k$ and perform the merge step through cross-worker communication. The result will be identical to when rendered on a single worker.

**Evaluating the condition of** $g_i \in N_{lk}$. Since the subspaces $S_1, S_2, \ldots, S_K$ seamlessly partition the entire space and each subspace is convex, it necessitates that all dividing surfaces are planar. Each subspace $S_k$ can be represented by a set of plane constraints: $S_k = \{x \in \mathbb{R}^3 : n_{km}^\top \cdot x + d_{km} \leq 0, \text{ for all } m\}$ , which allows us to transform the indicator function $\mathbb{1}$ in Eq 3 into a form more amenable to computation:

$$\mathbb{1}(g_i \in N_{lk}) = \mathbb{1}(x_i \in S_k) = \prod_m \mathbb{1}(n_{km}^\top \cdot x_i + d_{km} \leq 0), \tag{6}$$

where $x_i$ represents the world coordinates of the intersection between the ray $l$ and the 2D Gaussian primitive $G_i'$. Expressing the ray $l$ as the equation $l(t) = o + td$, which is defined by the camera center $o \in \mathbb{R}^3$ and the unit ray direction $d \in \mathbb{R}^3$, then $x_i$ precisely equals the projection center point $u_i$ of $g_i$ onto $l$, which can be represented as: $x_i = u_i + (d^\top \cdot (o - u_i))d$.

### 3.3 DISTRIBUTED TRAINING WITH SUB-MODELS

To balance each subset's size and each worker's workload, we employ a KD tree to determine the partition of the subspaces $\{S_k\}$. Initially, we construct a three-dimensional KD tree with a depth of $L$ using the center coordinates of Gaussian primitives. The KD tree recursively uses hyperplanes perpendicular to the X, Y, and Z axes to bisect the space and equally divided primitives, ultimately resulting in a series of rectangular subspaces $S_1, S_2, \ldots, S_K$, where $K = 2^L$. Then, the corresponding subsets for each worker can be derived as: $N_k = \{i : n_{km}^\top \cdot u_i + d_{km} \leq D_i, \text{ for all } m\}$, where $u_i$ is the center of primitives and $D_i$ is the truncation threshold for $g_i$. In our implementation, we set $D_i$ to be three times the length of the major axis of the Gaussian ellipsoid. Appendix.Fig 11 provides an intuitive illustration of how the primitives near the subspace boundary work.

### 3.4 THE COMPLETE TRAINING PIPELINE

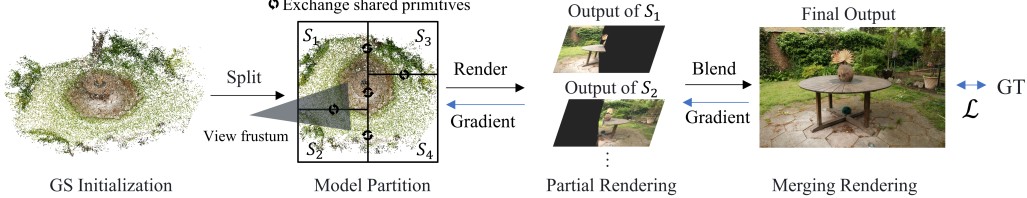

Figure 3: By employing planes generated using KD-Tree, we spatially partitioned the initial 3DGS model to a set of sub-models. These sub-models share certain primitives only when these primitives cross boundaries. The rendering results of sub-models are then merged to form the final rendered image. After the loss is computed uniformly, the corresponding gradients are returned to each sub-model to update their primitive parameters.

The actual training pipeline is shown in Figure 3. We denote each subset as a sub-model and assign it to a separate GPU, while a central manager is responsible for managing the KD-Tree and

the subspaces $\{S_n\}$. This manager also handles the parsing of incoming rendering requests and distributes rendering tasks to the relevant sub-models. The computational results from all sub-models, $\{T_k\}$ and $\{C_k\}$, are sent back to the central manager. These results can be represented in a 2D map with the same resolution as the target image, which consumes only minimal communication bandwidth. The central manager then completes the final rendering based on Eq 5, calculates the loss, and sends the gradient back to each sub-model for model parameter updates. After a predetermined number of training epochs, we repeat the partitioning process to accommodate any significant shift of primitive centers. Note that primitives belonging to intersections of multiple subsets require gradient synchronization after each training step. Empirically, we found this could be omitted without significantly affecting reconstruction quality.

### 3.4.1 PRIMITIVE INITIALIZATION

In the original 3D Gaussian Splatting (3DGS) approach, initialization and densification processes involved numerous heuristic strategies and hyperparameters to ensure that Gaussian primitives were appropriately positioned. Such strategies made it difficult to control the number of primitives, impeding our ability to effectively scale the model size and leverage the advantages of distributed modeling to enhance rendering fidelity. Furthermore, we found that the densification process does not always promote an increase in primitives in high-resolution training, which makes it counterproductive.

In this paper, we adopt a simple yet effective strategy for initialization as shown in Fig 3 and do not adjust the number afterwards. We performed Multi-View Stereo (MVS) on all training data to obtain depth estimations for all training viewpoints, which were then transformed into dense 3D point clouds. These point clouds could be flexibly sampled and initialized as Gaussian primitives. This streamlined approach allowed us to control the number of primitives as desired. It is also easier to balance workers' workloads thanks to a predefined number of primitives.

## 4 EXPERIMENTS

**Datasets.** Our performance evaluation spanned 4 datasets that comprising indoor and outdoor scenes. These datasets include all scenes from full-resolution *MipNeRF-360* (Barron et al., 2022). We extended our analysis to high-resolution scenarios based on the ScanNet++ dataset (Yeshwanth et al., 2023), with a focus on scenes labeled 108ec0b806 and 8133208cb6. After distortion correction, two scenes provided 863 and 476 high-resolution images respectively (8408 pixels wide). Furthermore, we conducted tests in large-scale environments, including the Residence, Building, and Rubble datasets from *Mega-NeRF* (Turki et al., 2022), as well as the entire *MatrixCity Small City Aerial* dataset (referred to as MatrixCity-Aerial, 1920×1080), consisting of 6,362 images. We also utilized the *MatrixCity Small City Dense Street* dataset, sampling 5 angles every meter along the centerline of streets to gather 135,290 images (1000×1000). From the *Small City Dense Street* dataset, we selected a focused test set of 2,480 images (referred to as MatrixCity-M). By combining all images from Aerial and Dense Street (referred to as MatrixCity-ALL), we conducted a comprehensive Billion GS level reconstruction. We followed the official Train/Test splits for *Mega-NeRF* and *MatrixCity-Aerial*, and used every eighth image for testing in other datasets as recommended by MipNeRF-360. To obtain superior MVS initial points, we reran Colmap's sparse reconstruction based on full-resolution images of MipNeRF-360 and ScanNet++ to obtain poses (using official provided poses of Mage-NeRF and MatrixCity), and subsequently performed dense reconstruction (Schönberger et al., 2016) on all datasets except MatrixCity-ALL. For MatrixCity-ALL, we replaced depth estimations of MVS with official provided depth estimations to avoid super long time cost of Colmap's dense reconstruction (using MVS for MatrixCity-M and MatrixCity-Aerial).

**Baselines and Metrics.** Our comparisons include 3DGS and NeRF-related works. For fair comparisons, results from an equal number of iterations from our own 3DGS runs are also presented. We primarily assessed the rendered image quality using three metrics consistent with 3DGS: PSNR, SSIM, and LPIPS.

**Implementation Details.** Our method is based on 3DGS. We extended the number of training iterations to 60k on MipNeRF-360, ScanNet++ and Maga-NeRF and 20 epochs on all MatrixCity datasets for both 3DGS and ours to ensure adequate convergence. We do not adjust the number of primitives during training. Since the primitives are initialized with relatively accurate position

parameters from MVS, we reduce the learning rate for the position parameters in all primitives from $1.6 \times 10^{-6}$ to $1.6 \times 10^{-8}$ with a exponential decay function, which is 1/100th of the original setting in 3DGS. All experiments are conducted on NVIDIA A100 GPUs.

## 4.1 EXPERIMENTAL RESULTS

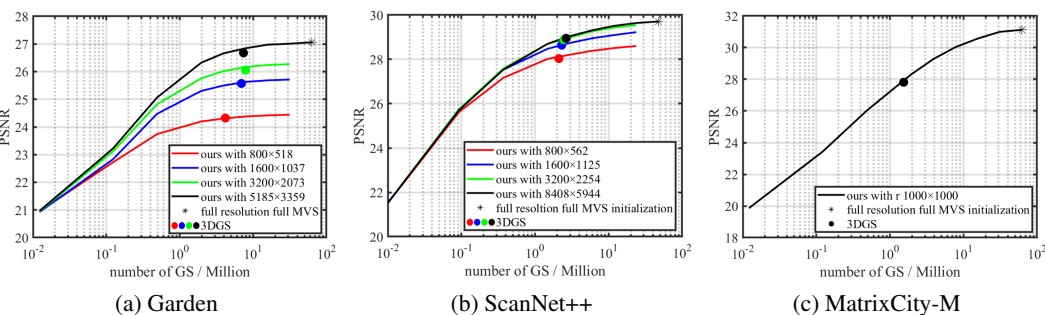

(a) Garden     (b) ScanNet++     (c) MatrixCity-M

Figure 4: **PSNR vs. #GS analysis on various datasets.** The markers represent 3DGS baseline, and the curves denote our method. Results at various training resolutions are upsampled and evaluated at the full resolution to unify the metrics. Our method achieves superior PSNR simply by naively scale the model size. This trend is more pronounced on the high-fidelity dataset ScanNet++ and larger-scale dataset MatrixCity-M.

Fig 4 presents our quantitative results on the MipNeRF-360 (Garden), ScanNet++ (108ec0b806), and MatrixCity-M with various training resolution. To standardize the metrics and align them more closely with direct visual perception, we upsample all rendered results from different resolutions to the full resolution using nearest-neighbor upsampling and evaluate on the full resolution. Utilizing MVS for initializing Gaussian points and disabling densification allows us to easily control the number of model points. Furthermore, our distributed training approach enables the use of a large number of Gaussian primitives. We observed a strong positive correlation between the number of Gaussian primitives and the final model's PSNR. When the number of primitives is similar, our PSNR closely matches the quality of 3DGS trained on a single GPU; however, we can effectively achieve higher PSNRs by simply increasing the number of primitives. Table 1 shows results on more datasets, uniformly showing that our method consistently performs better across all datasets, especially on high-resolution and large-scale datasets. Note that, although current state-of-the-art methods on bird-view datasets utilize complex post-processing or ensemble strategies, we achieved comparable results simply by increasing the number of Gaussian splats.

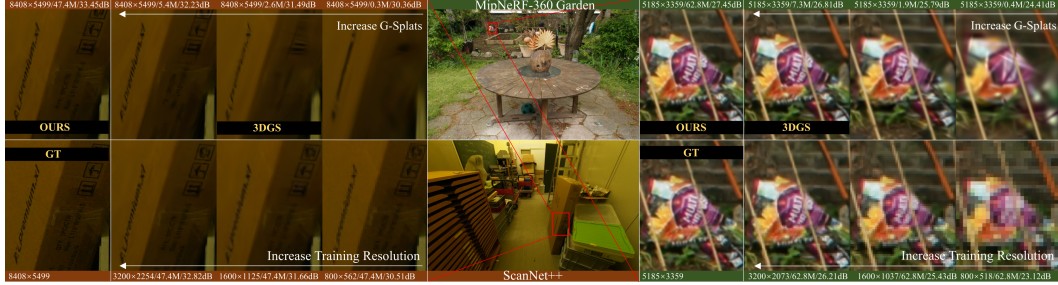

Figure 5: Visualization of models and with various number of primitives and training resolution on Garden and ScanNet++ dataset (top and bottom metrics: training resolution/splats count/PSNR). As we get close to objects or zoom into camera, higher training resolutions and more primitives help maintain rendering clarity and reveal more details, which bring better visual experience and better quantitative results than the 3DGS baseline.

We further analyze the relationship between the number of Gaussian points and subjective visual effects, as illustrated in Fig. 5. As revealed in Fig. 4, at a fixed number of GS primitives, higher resolutions yield better image quality; similarly, at a fixed high resolution, an increased number of primitives enhances image quality. We believe that the number of primitives determines the

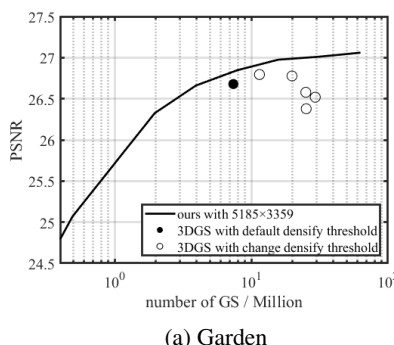 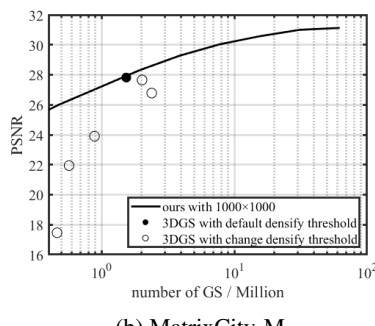

(a) Garden        (b) MatrixCity-M

Figure 6: **Comparative Analysis of Desification Strategies.** The curves denote our MVS initialization, showing a clear positive correlation between the number of primitives and visual quality. Baseline results from the 3DGS method with its default densification threshold of 0.0002 are marked as black dots. Lower thresholds were tested to assess their impact on point densification compared to the default setting. The findings suggest that reducing the threshold does not consistently increase model size and fails to outperform our method.

capacity of the 3DGS model, while higher image resolutions bring a greater amount of information, necessitating a larger model capacity to achieve adequate fitting. Therefore, to ensure a **Retina** quality effect, both high resolution and high-quality rendering are essential, which in turn imposes substantial demands on the number of GS primitives.

Table 1: Comparison with SOTA. RetinaGS is the first distributed method for general scenes, e.g. street and indoor scenes. It is also comparable to specialized methods for bird-view scenes via simply increasing splats. * means that results are averaged across scenes. See Appendix for complete results.

| Datasets
Pixels
Type | MatrixCity-Aerial
13.19B
**Bird-View** | | MatrixCity-M
2.48B
**Street-View** | | Mega-NeRF*
2.25B
**Bird-View** | | ScanNet++*
33.44B
**Indoor** | | MipNeRF-360*
2.32B
**Indoor** | |
|---|---|---|---|---|---|---|---|---|---|---|
| Metrics | PSNR | #GS | PSNR | #GS | PSNR | #GS | PSNR | #GS | PSNR | #GS |
| GP-NeRF | 23.56 | N/A | - | - | 22.46 | N/A | - | - | - | - |
| 3DGS | 23.67 | 9.7M | 27.62 | 1.01M | 22.56 | 6.57M | 28.42 | 1.87M | 27.33 | 3.02M |
| CityGaussian | 27.46 | 23.7M | - | - | **23.10** | 11.23M | - | - | - | - |
| Ours | **27.70** | 217.30M | **31.12** | 62.18M | 23.03 | 35.57M | **28.91** | 39.89M | **27.78** | 27.79M |

## 4.2 Exploration Study

**Initialization and Densification**. In practice, we observed issues with the original 3DGS's point-growing strategy, as shown in Appendix.Fig 12. Excessive iterations lead to deteriorating results. Using an aggressive densification strategy in 3DGS did not yield better outcomes, and excessive point splitting made the training unstable. In Fig 6, we show that initialization via MVS results in a more stable training run and better model quality. By simply initializing using MVS with more primitives, our model surpasses the original 3DGS even with careful tuning.

**Validity of Distribute Rendering** we devise a simple test to validate the correctness of the underlying rendering equation in our method. We precisely positioned the camera's optical axis on the dividing plane between two subspaces sharing this plane, ensuring that each ray passes through only one subspace. In this case, each image pixel should only be rendered by primitives on one side of the plane. If we do not perform the step, we would expect a crisp boundary between color pixels and completely dark pixels on the partially rendered images from the two subspaces. As shown in Fig. 7, our method exhibits the expected behavior while baseline partition approaches (Lin et al., 2024; Liu et al., 2024) that deviates from the proper rendering equation fails the test.

**KD-Tree Partition vs. Fixed-size Partition.** We compare our KD-Tree partitioning approach to a naive grid division strategy that uniformly divides the scene into blocks of fixed size in Table 2. The

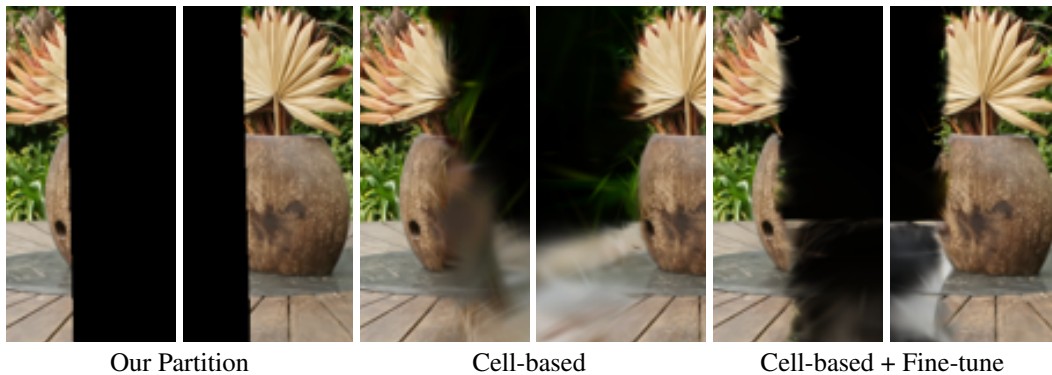

| Our Partition | Cell-based | Cell-based + Fine-tune |
|:---:|:---:|:---:|
| PSNR: 27.053 | PSNR: - | PSNR: 26.969 |

Figure 7: Our two submodels produced distinctly bounded outputs, and result in better PSNR results. In contrast, spatial-based partitions exhibited blurry boundaries that could not be entirely eliminated by comprehensive end-to-end refinement, demonstrating the superiority of our approach.

KD-Tree effectively balances the number of primitives in each sub-model, resulting in an optimal balance in peak memory usage and the best training speed. KD-tree is used throughout all experiments work without further notice.

Table 2: **Partition efficiency.** Range of primitive count, memory usage, total time, and communication time of submodels under different partition strategies in distributed training. The fixed-size division resulted in sub-models with significantly varied number of primitives and computation times, which increased time spent blocking on waiting, thereby reducing the overall system efficiency.

| Partition | Batch Size | GPU | Primitives (M) | Mem.(GiB) | Total Time (s) | Comm. (s) |
|---|---|---|---|---|---|---|
| KD-Tree | 2 | 2 | 2.87~2.88 | 6.26~6.32 | 26.74~26.76 | 4.39~6.04 |
| Fixed-size | 2 | 2 | 0.81~4.93 | 3.57~9.01 | 32.45~32.52 | 1.04~23.83 |
| KD-Tree | 4 | 4 | 1.45~1.47 | 5.26~5.40 | 18.87~18.90 | 5.33~7.34 |
| Fixed-size | 4 | 4 | 0.05~4.79 | 3.08~10.95 | 32.26~32.41 | 4.77~26.83 |

**Model parallelism vs. Data parallelism.** We compare data parallel training and training with our method, which is model parallelism, on the MipNeRF-360-garden dataset. As shown in Table 3, our method achieve lower peak memory usage and higher training throughput. It is evident that even with such a small GS model, our method maintains its advantages over DP and single GPU training. However, as the size of submodels decreases and their quantity increases, it becomes increasingly challenging for MP to achieve workload balance. As shown in Table 4, a straightforward solution is to increase the batch size, which statistically leads to a more balanced workload.

Table 3: **Parallel mechanism efficiency.** Efficiency of data parallel(DP) and our model parallel(MP) with various setting.

| Parallel | GPU | Batch Size | Mem. | Time | Comm. |
|---|---|---|---|---|---|
| - | 1 | 1 | 8.33 | 32.35 | 0 |
| DP | 2 | 2 | 9.66 | 45.36 | 29.22 |
| MP | 2 | 2 | **6.26** | **26.76** | **6.038** |
| DP | 4 | 4 | 9.66 | 30.62 | 21.56 |
| MP | 4 | 4 | **5.26** | **18.90** | **7.34** |
| DP | 8 | 8 | 9.66 | 20.38 | 16.51 |
| MP | 8 | 8 | **5.05** | **17.738** | **12.73** |

Table 4: **Efficiency vs. Batch Size.** Increasing the batch size enhances the balance of computation across processes, resulting in reduced communication blocking.

| Bacth Size | GPU-ID | Mem. | Time | Comm. |
|---|---|---|---|---|
| 1 | 0 | 5.52 | 31.20 | 6.51 |
| | 1 | 5.31 | 31.84 | 10.94 |
| 2 | 0 | 6.26 | 26.76 | 6.04 |
| | 1 | 6.32 | 26.74 | 4.39 |
| 4 | 0 | 7.79 | 25.29 | 6.20 |
| | 1 | 7.93 | 25.30 | 3.02 |

## 5 TRAINING BILLION-SCALE 3DGS

Our distributed modeling method enables the GS model to scale up to extremely large sizes. Typically, scenes in previous work are trained on a single GPU using 10 to 200 million pixels with a GS model with **millions** of primitives. In this work, we use 64 A100 GPUs for 10 days to train a 3DGS model with a **billion** primitives on the MatrixCity-ALL dataset, which includes **141,652 images** containing over **148 billion** pixels. There have been no previous reports of successful training at this scale of training set or model size. After training for 20 epochs, billion-scale model achieve superior results compared to million-scale model as shown in Fig 8, Table. 5 and **supplementary videos**.

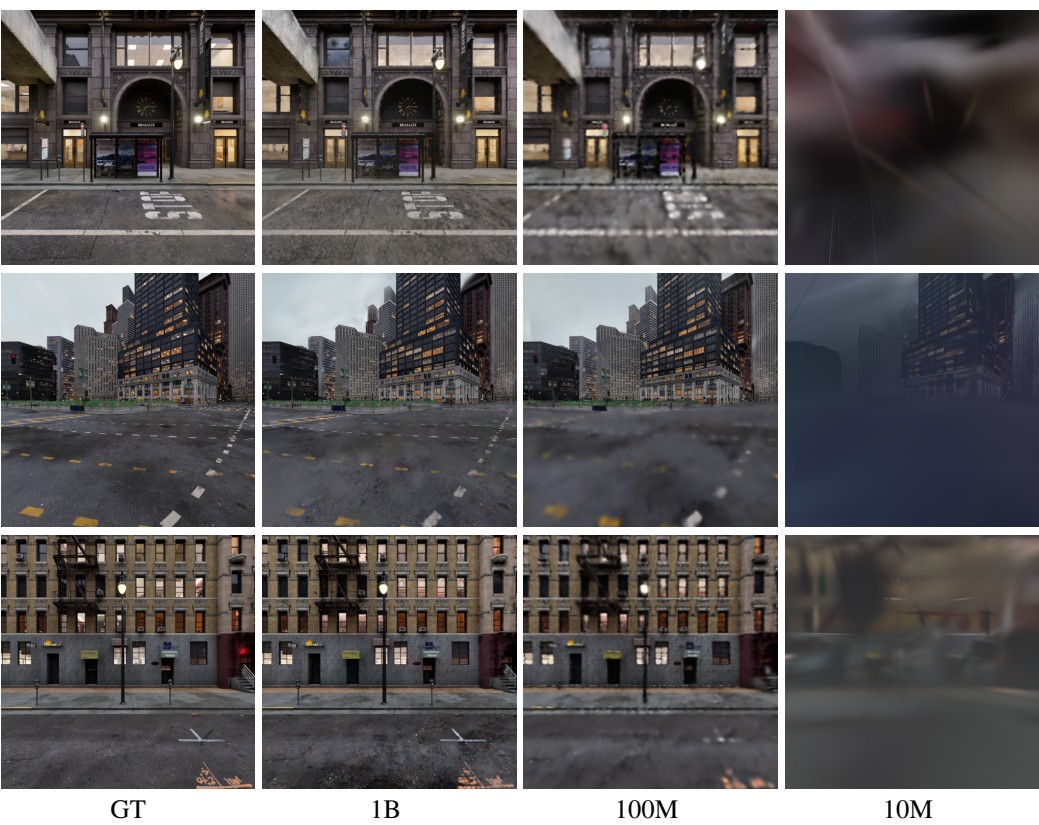

GT        1B        100M        10M

Figure 8: 1 billion vs. 100 million vs. 10 million splats for Maxtricity-ALL reconstruction.

Table 5: Qualitative results on full MatrixCity-All dataset (with 148 billion pixels). More primitives effectively improve performance with the help of the proposed distributed training method, reaching quality unattainable by single GPU training.

| Metrics | SSIM↑ | PSNR↑ | LPIPS↓ | #GS | GPU |
|---------|-------|-------|--------|-----|-----|
| Ours-10M | 0.608 | 16.53 | 0.536 | 10.00M | 1 |
| Ours-100M | 0.761 | 23.07 | 0.397 | 100.00M | 8 |
| Ours-1B | 0.815 | 25.50 | 0.282 | 1023.13M | 64 |

## 6 CONCLUSION AND LIMITATION

In this paper, we study the problem of distributed training of 3DGS models. We devise a model parallelism-based training method that utilizes a proper rendering equation to avoid artifacts. This allows us to significantly expand the model scale in terms of primitive numbers and seamlessly support large-scale scene reconstruction and detailed rendering. Although our method allows for improving the model's capacity by simply increasing the number of splats, this also results in a higher computational load per ray. We believe that an effective hierarchical Level-of-Detail (LOD) description will address this issue, leading to enhancements in performance and rendering quality.

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

# A   APPENDIX

## A.1   MORE VISUALIZATIONS

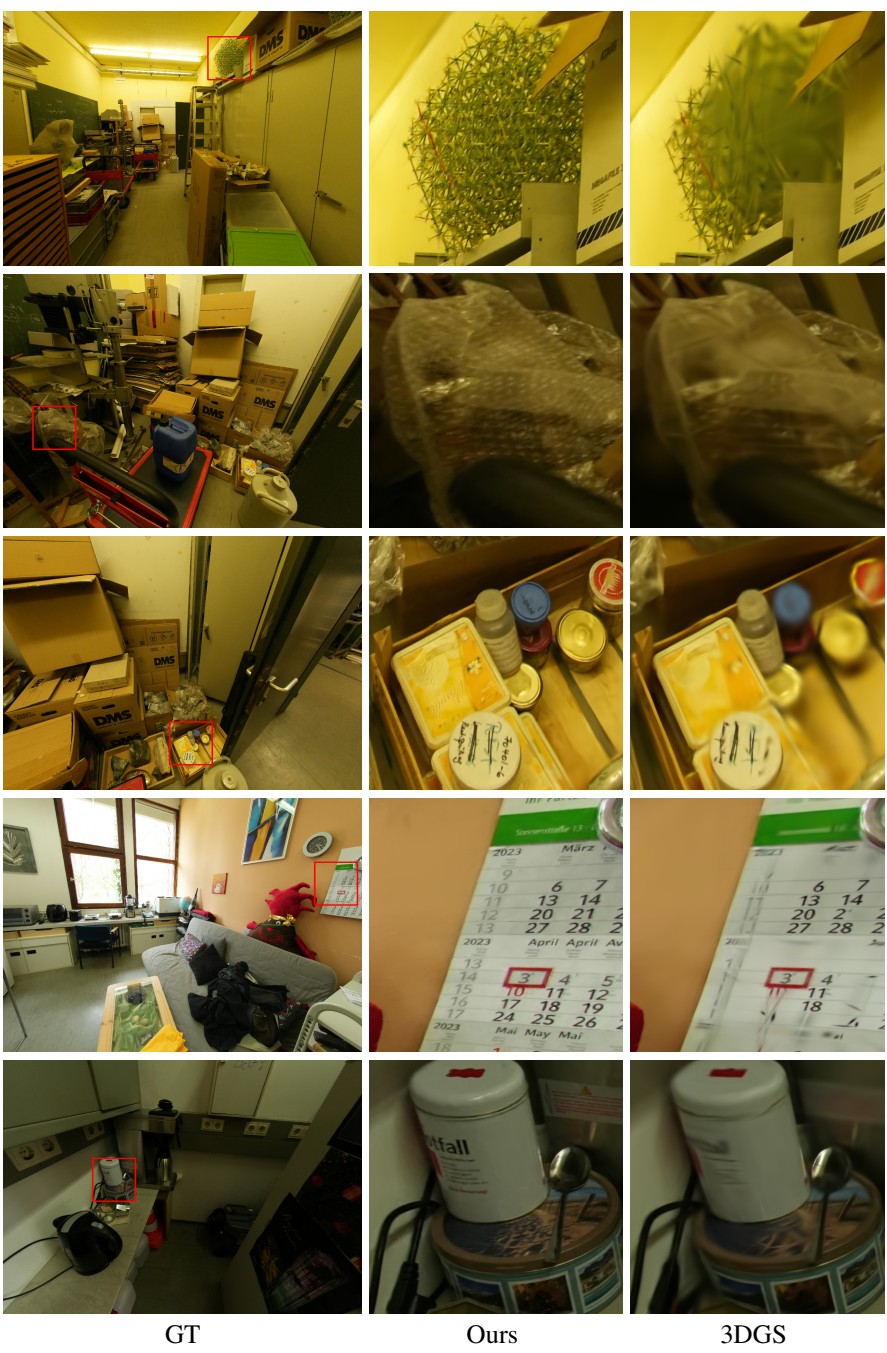

GT                    Ours                    3DGS

Figure 9: Comparisons of high-primitive-number RetinaGS models and 3DGS baseline on ScanNet++ dataset. Note the superior rendering quality, especially on high-frequency textures like text and leaves.

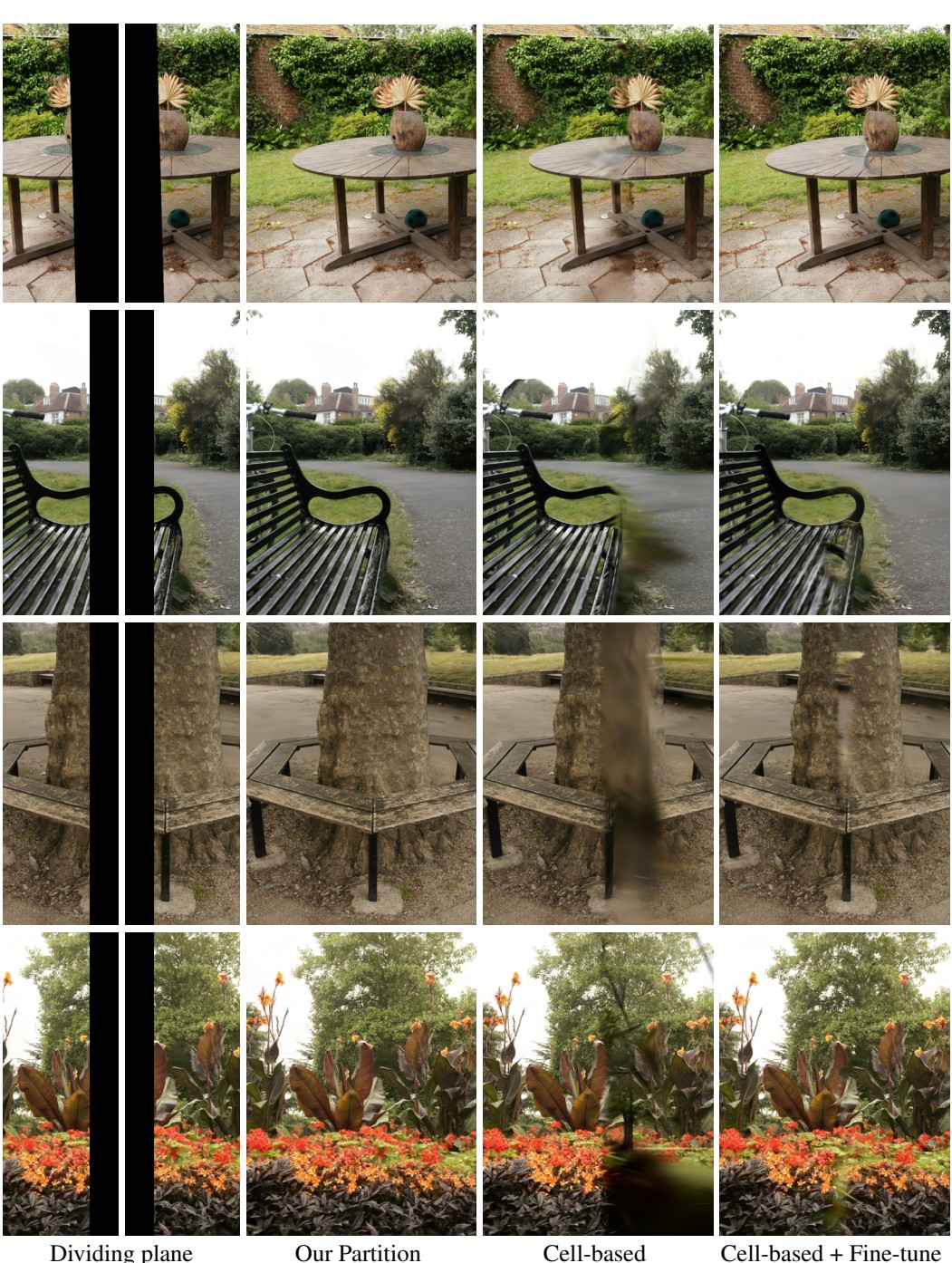

|  |  |  |  |
|---|---|---|---|
| Dividing plane | Our Partition | Cell-based | Cell-based + Fine-tune |

Figure 10: Blending results at the sub-space boundaries for different partition approaches. Approximate methods leak through the boundaries in partial rendering, resulting in obvious artifacts after merging. RetinaGS uses the equivalent form the 3DGS rendering equation, so it does not have this issue and shows no artifacts in the final rendering.

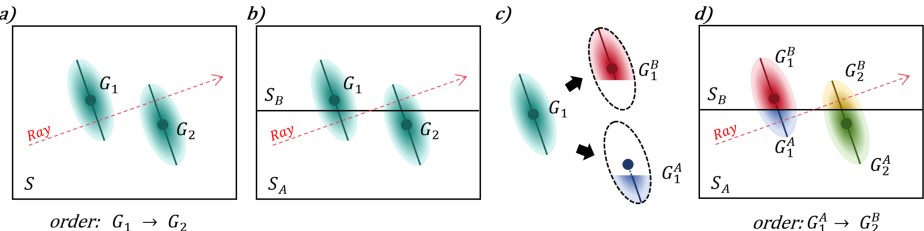

Figure 11: In RetinaGS, one primitive may be involved in partial value computation of more than one overlapping subsets, but the indicator function in Eq 6 ensures sure its color and opacity will only take effect once. For each ray, it can be illustrated as each subspace only possesses a fraction of the Gaussian ellipsoid, and the fragments collaboratively and distributively accomplish the task of a single primitive.

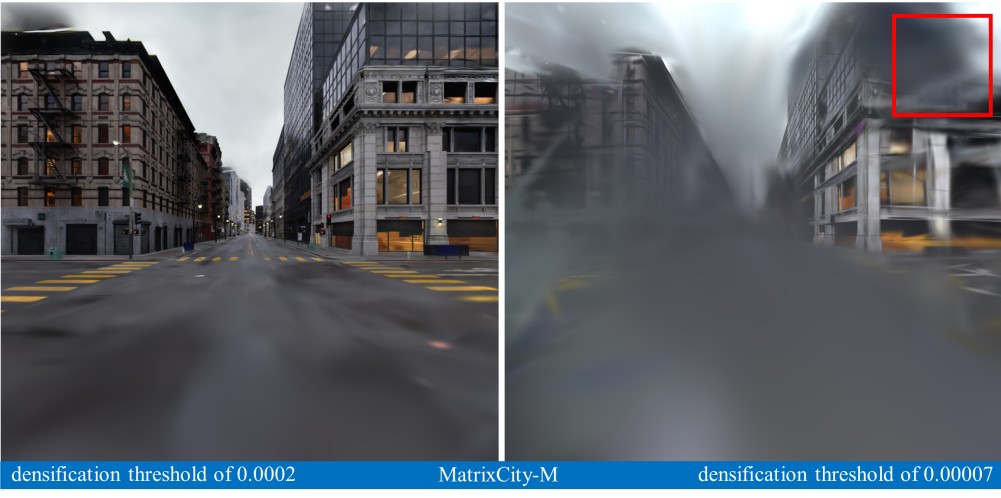

Figure 12: In 3DGS, as the densification threshold decreases, the growth rate of GS increases. However, due to the rapid growth rate, floaters are generated in the model that cannot be eliminated through training (marked with red rectangle), which deteriorates the reconstruction effect. The MVS initialization strategy introduced in RetinaGS will mitigate this issue.

## A.2 EQUIVALENCE DERIVATION

Starting from Eq. 3 for partial color $C_k(l)$:

$$C_k(l) = \sum_{g_i \in N_{lk}} c_i \sigma_i \prod_{g_j \in N_{lk},\, j<i} (1 - \sigma_j),$$

Eq. 4 for partial opacity $T_k(l)$:

$$T_k(l) = \prod_{g_i \in N_{lk}} (1 - \sigma_i),$$

and Eq. 5 for merging formula:

$$C(l) = \sum_{k \in o_l} C_k(l) \prod_{m \in o_l,\, m<k} T_m(l),$$

we substitute the expressions for partial color and partial opacity into the merging formula:

$$C(l) = \sum_{k \in o_l} \left( \sum_{g_i \in N_{lk}} c_i \sigma_i \prod_{g_j \in N_{lk},\, j<i} (1 - \sigma_j) \right) \prod_{m \in o_l,\, m<k} \left( \prod_{g_j \in N_{lm}} (1 - \sigma_j) \right).$$

Now we expand this expression step by step. First, applying the distributive property of multiplication, we can pull out the inner summation symbol:

$$C(l) = \sum_{k \in o_l} \sum_{g_i \in N_{lk}} c_i \sigma_i \left( \prod_{g_j \in N_{lk},\, j<i} (1 - \sigma_j) \right) \left( \prod_{m \in o_l,\, m<k} \prod_{g_j \in N_{lm}} (1 - \sigma_j) \right).$$

Then, utilizing the commutative property of multiplication, we can rewrite the above equation as:

$$C(l) = \sum_{k \in o_l} \sum_{g_i \in N_{lk}} c_i \sigma_i \left( \prod_{g_j \in N_{l1}} (1 - \sigma_j) \right) ... \left( \prod_{g_j \in N_{l(k-1)}} (1 - \sigma_j) \right) \cdot \left( \prod_{g_j \in N_{lk},\, j<i} (1 - \sigma_j) \right).$$

Since $N_l$ in Eq 2 and $N_{lk}$ in Eq 3 follow a consistent rule for the ordering of elements, we have $N_l = (N_{l1}, N_{l2}, ...N_{lK})$. This allows us to reduce the product symbols to one:

$$C(l) = \sum_{k \in o_l} \sum_{g_i \in N_{lk}} c_i \sigma_i \left( \prod_{g_j \in N_l,\, j<i} (1 - \sigma_j) \right).$$

Similarly, nested summations can also be simplified into one:

$$C(l) = \sum_{g_i \in N_l} c_i \sigma_i \left( \prod_{g_j \in N_l,\, j<i} (1 - \sigma_j) \right),$$

which has the exact same form as Eq 2. Now we prove the original rendering equation of 3DGS is equivalent to the hierarchical form underlying the distributed training framework of RetinaGS.

## A.3 IMPLEMENTATION DETAILS OF TRAINING BILLION-SCALE 3DGS

For the initialization of primitives, we employed depth maps officially provided by MatrixCity, where each aerial view image contributes approximately 1/64 million primitives, and each street view image contributes about 1/128 million primitives (excluding pixels exceeding the maximum depth). Due to the application of anti-aliasing on the depth maps, the directly generated initialization points are accompanied by noise. Such noise in the initialization can lead to floters in the training results, potentially compromising the visual quality of the model. To obtain a clean point cloud, we utilized DBSCAN for noise filtration, treating aerial and street views separately. For aerial views, the epsilon

and minimum number of points were set to 1 m and 15, respectively, while for street views, these parameters were established at 1 m and 20, respectively. Following the noise reduction process, an additional 2% of primitives were added to represent the sky (initialized as hemispheres at a fixed distance from the city center, with the fixed distance being twice the length of the city), culminating in a total of 1 Billion primitives.

We spatially partitioned the initial primitives to 64 sub-models and do not adjust the number of primitives during training. Since the primitives are initialized with accurate geometrical parameters, we reduce the learning rate for the position parameters from $1.6 \times 10^{-7}$ to $1.6 \times 10^{-9}$ with a exponential decay function. Besides, we reduce the learning rate for the scale parameters from $5 \times 10^{-4}$ to $5 \times 10^{-6}$ with a exponential decay function to avoid unusually large primitives.

## A.4 MORE EXPERIMENTAL RESULTS

In the main text, for the sake of brevity, we did not present the experimental results for each individual scene on several datasets. Instead, we provided the average results across all scenes contained in each dataset (as highlighted with * in the experimental tables of the main text). Here, we present the detailed experimental data for each individual scene.

Table 6: Experimental results on MatrixCity. R & P means resolution and training pixels. The dagger †indicates that the result was obtained using the same number of training iterations (20 epochs) as our method.

| Datasets
R & P | MatrixCity-Aerial
1920×1080 & 13.19B | | | | MatrixCity-M
1000×1000 & 2.48B | | | |
|---|---|---|---|---|---|---|---|---|
| Metrics | SSIM↑ | PSNR↑ | LPIPS↓ | #GS | SSIM↑ | PSNR↑ | LPIPS↓ | #GS |
| 3DGS | 0.735 | 23.67 | 0.384 | 9.70M | 0.839 | 27.62 | 0.282 | 1.01M |
| CityGaussian | **0.865** | 27.46 | 0.204 | 23.70M | - | - | - | - |
| 3D-GS† | 0.833 | 26.56 | 0.244 | 25.06M | 0.851 | 27.81 | 0.271 | 1.53M |
| Ours | 0.840 | **27.70** | **0.177** | 217.3M | **0.932** | **31.12** | **0.110** | 62.18M |

Table 7: Experimental results (60K iterations) on Mega-NeRF.

| Datasets | Residence | | | Rubble | | | Building | | |
|---|---|---|---|---|---|---|---|---|---|
| Metrics | SSIM↑ | PSNR↑ | LPIPS↓ | SSIM↑ | PSNR↑ | LPIPS↓ | SSIM↑ | PSNR↑ | LPIPS↓ |
| Mega-NeRF | 0.628 | 22.08 | 0.489 | 0.553 | 24.06 | 0.516 | 0.547 | 20.93 | 0.504 |
| Switch-NeRF | 0.654 | **22.57** | 0.457 | 0.562 | 24.31 | 0.496 | 0.579 | 21.54 | 0.474 |
| GP-NeRF | 0.661 | 22.31 | 0.448 | 0.565 | 24.06 | 0.496 | 0.566 | 21.03 | 0.486 |
| 3DGS | 0.751 | 21.43 | 0.274 | 0.709 | 24.47 | 0.337 | 0.723 | 21.74 | 0.302 |
| CityGaussian | **0.813** | 22.00 | **0.211** | **0.813** | **25.77** | **0.228** | **0.778** | 21.55 | 0.246 |
| Ours | 0.781 | 21.87 | 0.217 | 0.760 | 25.09 | 0.234 | 0.754 | **22.14** | **0.227** |

Table 8: PSNR vs. Primitive numbers (60K iterations) on Mega-NeRF.

| Datasets
R & P | Residence
1368×912 & 3.19B | | Rubble
1152×864 & 1.64B | | Building
1152×864 & 1.91B | |
|---|---|---|---|---|---|---|
| Metrics | PSNR | #GS | PSNR | #GS | PSNR | #GS |
| 3DGS | 21.43 | 6.42M | 24.47 | 4.7M | 21.44 | 8.6M |
| Ours | **21.87** | 51.41M | **25.09** | 27.9M | **22.14** | 27.4M |

Table 9: Experimental results (60K iterations) on ScanNet++.

| Datasets | 108ec0b806 | | | | 8133208cb6 | | | |
| R & P | 8408×5944 & 43.13B | | | | 8408×5935 & 23.75B | | | |
| Metrics | SSIM↑ | PSNR↑ | LPIPS↓ | #GS | SSIM↑ | PSNR↑ | LPIPS↓ | #GS |
|---|---|---|---|---|---|---|---|---|
| 3DGS | 0.881 | 28.95 | 0.408 | 2.65M | **0.908** | 27.89 | 0.355 | 1.09M |
| Ours | **0.883** | **29.71** | **0.395** | 47.59M | **0.908** | **28.11** | **0.340** | 32.19M |

Table 10: #GS for Mip-NeRF360 scenes under 1.6K resolution and full resolution. The full marks the results obtained in full resolution. The dagger †marks the results obtained in our own experiments (60K iterations).

| Method | Scenes | bicycle | garden | flowers | stump | treehill | room | counter | kitchen | bonsai |
| Resolution-Wide | | 4946 | 5187 | 5025 | 4978 | 5068 | 3114 | 3115 | 3115 | 3118 |
| Pixels-full | | 3.15B | 3.22B | 2.87B | 2.05B | 2.37B | 2.00B | 1.55B | 1.80B | 1.89B |
|---|---|---|---|---|---|---|---|---|---|---|
| 3D-GS† | | 7.03M | 6.92M | 4.11M | 5.34M | 4.17M | 1.80M | 1.27M | 1.95M | 1.05M |
| Ours | | 31.67M | 62.94M | 20.53M | 15.32M | 22.75M | 22.41M | 22.83M | 28.31M | 23.43M |
| 3D-GS†-full | | 5.04M | 7.39M | 3.11M | 3.30M | 2.61M | 1.70M | 1.20M | 1.91M | 0.97M |
| Ours-full | | 31.67M | 62.94M | 20.53M | 15.32M | 22.75M | 22.41M | 22.83M | 28.31M | 23.43M |

Table 11: SSIM scores for Mip-NeRF360 scenes under 1.6K resolution and full resolution. The full marks the results obtained in full resolution. The dagger †marks the results obtained in our own experiments (60K iterations).

| Method | Scenes | bicycle | garden | flowers | stump | treehill | room | counter | kitchen | bonsai |
|---|---|---|---|---|---|---|---|---|---|---|
| 3D-GS | | 0.771 | 0.868 | 0.605 | 0.775 | 0.638 | 0.914 | 0.905 | 0.922 | 0.938 |
| 3D-GS† | | 0.770 | 0.866 | 0.623 | 0.771 | 0.641 | 0.931 | 0.919 | 0.933 | 0.950 |
| Mip-NeRF360 | | 0.685 | 0.813 | 0.583 | 0.744 | 0.632 | 0.913 | 0.894 | 0.920 | 0.941 |
| iNPG | | 0.491 | 0.649 | 0.450 | 0.574 | 0.518 | 0.855 | 0.798 | 0.818 | 0.890 |
| Plenoxel | | 0.496 | 0.606 | 0.431 | 0.523 | 0.509 | 0.841 | 0.759 | 0.648 | 0.814 |
| Ours | | **0.776** | **0.870** | **0.624** | **0.777** | **0.650** | **0.935** | **0.927** | **0.941** | **0.958** |
| 3D-GS†-full | | 0.732 | 0.817 | 0.604 | 0.795 | 0.696 | 0.922 | 0.917 | 0.926 | 0.941 |
| Ours-full | | **0.751** | **0.827** | **0.635** | **0.800** | **0.703** | **0.927** | **0.925** | **0.932** | **0.947** |

Table 12: PSNR scores for Mip-NeRF360 scenes under 1.6K resolution and full resolution. The full marks the results obtained in full resolution. The dagger †marks the results obtained in our own experiments (60K iterations).

| Method | Scenes | bicycle | garden | flowers | stump | treehill | room | counter | kitchen | bonsai |
|---|---|---|---|---|---|---|---|---|---|---|
| 3D-GS | | 25.25 | 27.41 | 21.52 | 26.55 | 22.49 | 30.63 | 28.70 | 30.32 | 31.98 |
| 3D-GS† | | 25.33 | 27.58 | 21.85 | 26.74 | 22.46 | 31.93 | 29.54 | 31.52 | 33.10 |
| Mip-NeRF360 | | 24.37 | 26.98 | 21.73 | 26.40 | **22.87** | 31.63 | 29.55 | 32.23 | 33.46 |
| iNPG | | 22.19 | 24.60 | 20.34 | 23.63 | 22.36 | 29.27 | 26.44 | 28.55 | 30.34 |
| Plenoxel | | 21.91 | 23.49 | 20.09 | 20.66 | 22.24 | 27.59 | 23.62 | 23.42 | 24.67 |
| Ours | | **25.41** | **27.74** | **21.94** | **26.86** | 22.67 | **32.86** | **29.91** | **32.49** | **34.09** |
| 3D-GS†-full | | 24.47 | 26.67 | 20.78 | 26.23 | 22.34 | 31.57 | 29.33 | 31.72 | 32.92 |
| Ours-full | | **24.86** | **27.06** | **21.58** | **26.58** | **22.42** | **31.65** | **29.90** | **32.23** | **33.76** |

Table 13: LPIPS scores for Mip-NeRF360 scenes under 1.6K resolution and full resolution. The full marks the results obtained in full resolution. The dagger †marks the results obtained in our own experiments (60K iterations).

| Method | Scenes | bicycle | garden | flowers | stump | treehill | room | counter | kitchen | bonsai |
|---|---|---|---|---|---|---|---|---|---|---|
| 3D-GS | | 0.205 | 0.103 | 0.336 | **0.210** | 0.317 | 0.220 | 0.204 | 0.129 | 0.205 |
| 3D-GS† | | 0.200 | 0.107 | 0.320 | 0.223 | 0.317 | 0.186 | 0.172 | 0.112 | 0.168 |
| Mip-NeRF360 | | 0.301 | 0.170 | 0.344 | 0.261 | 0.339 | 0.211 | 0.204 | 0.127 | 0.176 |
| iNPG | | 0.487 | 0.312 | 0.481 | 0.450 | 0.489 | 0.301 | 0.342 | 0.254 | 0.227 |
| Plenoxel | | 0.506 | 0.386 | 0.521 | 0.503 | 0.540 | 0.418 | 0.441 | 0.447 | 0.398 |
| Ours | | **0.181** | **0.100** | **0.316** | 0.218 | **0.287** | **0.160** | **0.139** | **0.099** | **0.134** |
| 3D-GS†-full | | 0.375 | 0.268 | 0.438 | 0.386 | 0.447 | 0.277 | 0.261 | 0.188 | 0.263 |
| Ours-full | | **0.290** | **0.195** | **0.379** | **0.334** | **0.362** | **0.233** | **0.212** | **0.163** | **0.223** |

