# OpenReview forum: "RetinaGS: Scalable Training for Dense Scene Rendering with Billion-Scale 3D Gaussians"
_ICLR.cc/2025/Conference — Submitted to ICLR 2025_

### Official Review · Reviewer_fbZX · 2024-10-28

**Soundness:** 2
**Presentation:** 1
**Contribution:** 2
**Rating:** 5
**Confidence:** 4

**Summary:**

This paper presents RetinaGS, which utilizes a subspace-based parallelization method as a learning approach for Gaussian Splatting. Each subspace is handled by a single process and rendered using a rendering function with indicators. Instead of the standard splitting algorithm in Gaussian Splatting, MVS primitives are used to control the total number of Gaussians.

**Strengths:**

1. RetinaGS is a 3DGS model capable of training on large-scale datasets through parallelization.

2. The subspace-level rendering pipeline parallelization using a KD-Tree is intriguing.

3. RetinaGS allows for a controllable number of Gaussians.

**Weaknesses:**

1. In Table 1, the number of Gaussians is more than five times that of 3DGS. In Figures 5 and 6, reducing the number of Gaussians to 1/10 lowers the PSNR by about 2. Compared to 3DGS, if RetinaGS has the same number of Gaussians, its PSNR would likely be even lower. Therefore, the proposed method does not appear effective relative to the increasing number of Gaussians.

2. It appears that RetinaGS depends on the accuracy of the MVS algorithm.

3. I recommend a comparison with Hierarchical3DGS[1], as both methods present chunk-based Gaussian Splatting approaches.

4. Missing a comparison of rendering speeds.

5. Fig. 5 is difficult to understand. First, the resolution/splat, count, and PSNR are not clearly visible. Additionally, the images without labels are ambiguous in meaning. For instance, the images in the bottom row all seem to represent GT.

[1] Kerbl et al., *A Hierarchical 3D Gaussian Representation for Real-Time Rendering of Very Large Datasets*, ACM Transactions on Graphics, 2024

**Questions:**

1. Does *k* represent the *k*-th index?

2. During rendering, is it necessary to explore all subspaces along the ray path? On average, how many subspaces does each ray hit?

$~$

Minor comments:

There is an inconsistency in figure references on line 269.

Dots are missing in abbreviations on lines 233, 252, 274, 347, 417, 419, and 879.

A dot is missing in the sentence "Validity of Distribute Rendering" on line 422.

---

> ### Author Response · Authors · 2024-11-27
>
> **Q1: ... the proposed method does not appear effective relative to the increasing number of Gaussians.**
>
> In fact, RetinaGS is equivalent to 3DGS in terms of training and rendering. In Fig. 4, we compare RetinaGS with 3DGS, and as can be seen, for the same number of splats, the performance of RetinaGS always closely matches that of 3DGS. Additionally, increasing the number of splats shows a significant improvement in PSNR for RetinaGS. Especially for the street scene dataset MatrixCity-M, we can observe that increasing the splats by a factor of 10 improved the PSNR from 27.8 to 30.8. (Note that PSNR is a logarithmic metric.)
>
> **Q2: It appears that RetinaGS depends on the accuracy of the MVS algorithm.**
>
> In fact, our primary purpose of using MVS in this paper is to obtain a sufficient number of Gaussian splats. We could rely solely on the original densification strategy for training, which would also converge to reasonable results. However, the original densification strategy does not allow Gaussian splats to scale to large sizes, an issue we have thoroughly discussed in Section 4.2 of the paper. In other words, we do not rely on the accuracy of MVS, but rather on the point cloud density that MVS provides. This is also evident in Fig. 6, where the black dots represent the results obtained from 3DGS native parameters, and RetinaGS shows only a slight improvement in metrics for the same number of splats. This demonstrates that the improvement from MVS accuracy is limited, and the real advantage of RetinaGS lies in its ability to scale the number of splats.
>
> **Q3: I recommend a comparison with Hierarchical3DGS, as both methods present chunk-based Gaussian Splatting approaches.**
>
> Thank you for pointing this out. We will include the mentioned works in the Related Work section and provide a detailed discussion. HierarchicalGS introduces the concept of Level of Detail (LOD) to 3DGS, significantly reducing the number of splats involved in rendering for each view and thereby improving rendering speed. There are two key distinctions to highlight here:
> 1. Focus on scaling directions: HierarchicalGS focuses on “scaling in” by reducing the computational load of a single sub-model. In contrast, RetinaGS focuses on “scaling out” by enabling parallel training of more sub-models, thereby expanding the system's scale. In large-scale models, such as LLMs, scale-in and scale-out strategies are not mutually exclusive. In fact, they should often be employed together to maximize system scalability. We appreciate the reviewer for raising this perspective.
> 2. Memory and computational costs: In HierarchicalGS, all splats are still maintained as leaf nodes of the hierarchical tree, and additional parent nodes must also be maintained. As a result, GPU memory requirements do not decrease but rather increase, potentially requiring more computational nodes to accommodate the same model scale. This highlights the necessity of introducing a scale-out solution like RetinaGS.
>
> **Q4: Missing a comparison of rendering speeds.**
>
> In Table 2, we list the total time for rendering and backward passes under the same setup (with the same 3DGS model and the same number of inference passes). Notably, RetinaGS benefits significantly from distributed rendering. For instance, with 5.74 million primitives, 3DGS renders at 80 FPS on one GPU. In comparison, RetinaGS achieves the same rendering results at 80 FPS on a single GPU, 127 FPS with two GPUs, and 177 FPS with four GPUs.
>
> **Q5: Fig. 5 is difficult to understand.**
>
> Thank you very much for your feedback. We will carefully improve this figure and update it in the revised manuscript.
>
> **Q6: Does k represent the k-th index?**
>
> In Eq. 3–5, k represents the index of the sub-model or sub-space.
>
> **Q7: During rendering, is it necessary to explore all subspaces along the ray path? On average, how many subspaces does each ray hit?**
>
> Yes, based on our current implementation. We conducted statistical analysis across different datasets, and the results are presented in Fig. 2 of the main text. For aerial view datasets, rays typically intersect with only one block, while for indoor datasets (e.g., ScanNet++), they intersect with an average of 2.5 blocks.

---

> ### Comment · Reviewer_fbZX · 2024-12-03
>
> I appreciate the author's comment. I will keep my score as I still have some concerns. Additional comments are as follows.
>
> **Q3**: As in the comment of reviewer oADR, Hierarchical3DGS requires consistent performance improvement to show the advantages of RetianGS, as there is a comparison with the real world dataset. I agree that this comparison experiment should be added.
>
> **Q4**: I understand that the rendering speed increases as the number of GPUs increases, but more comparison with other methodologies is needed. For example, in the case of Hierarchical3DGS, I know that the FPS can go up to 159 depending on the optimization level. I think that one example is not enough for comparisons.

---

> > ### Author Response · Authors · 2024-12-04
> >
> > **Q3: As in the comment of reviewer oADR, Hierarchical3DGS requires consistent performance improvement to show the advantages of RetianGS, as there is a comparison with the real world dataset. I agree that this comparison experiment should be added.**
> >
> > Most of the experiments in HierarchicalGS were conducted on a custom dataset consisting of four street-view scenes created by the authors. Currently, only the smallest scene, named SmallCity, has been released, which contains only a few thousand images. Additionally, they reported experimental results on publicly available datasets of a similar scale (Building and Rubble scenes provided by Mega-NeRF). We chose these datasets for our comparative experiments. The experimental results are as follows:
> >
> > | Method                                   | Building | Rubble |
> > |------------------------------------------|----------|--------|
> > | RetinaGS + MVS                           | **22.14**    | **25.77**  |
> > | RetinaGS + sparse initialization         | **21.83**    | **24.85**  |
> > | HierarchicalGS + sparse initialization + depth prior   | 21.52    | 24.64  |
> >
> > Even with sparse initialization, RetinaGS achieves better PSNR. It is worth noting that HierarchicalGS additionally uses depth information as supervision. According to the results reported by HierarchicalGS, rendering with only leaf nodes consistently produces better visual quality compared to using the hierarchy. When the number of leaf nodes is the same, the hierarchy only provides faster rendering and training speeds but at the cost of reduced rendering quality and increased memory usage.
> >
> > **Q4: I understand that the rendering speed increases as the number of GPUs increases, but more comparison with other methodologies is needed. For example, in the case of Hierarchical3DGS, I know that the FPS can go up to 159 depending on the optimization level. I think that one example is not enough for comparisons.**
> >
> > Rendering speed is an advantage of HierarchicalGS. Under the same number of points, our method is almost equivalent to HierarchicalGS using only leaf nodes (58 FPS vs.58 FPS). Its rendering speed is slower compared to HierarchicalGS configured with the maximum granularity settings (58 FPS vs.159 FPS).

---

### Official Review · Reviewer_aDvg · 2024-10-31

**Soundness:** 3
**Presentation:** 3
**Contribution:** 3
**Rating:** 6
**Confidence:** 4

**Summary:**

The authors aim to achieve exceptional rendering visual fidelity, which requires training GS models with higher spatial resolution, larger datasets, and diverse viewing perspectives. However, current 3DGS training struggles in these settings. A core issue identified is that 3DGS training remains constrained by single-GPU setups, which become infeasible for handling even moderately sized scenes due to time and memory demands. While recent distributed training approaches split the scene data into subspaces processed independently on multiple GPUs, they rely on fixed data layouts like bird's-eye views that minimize ray overlap between subspaces. This partitioning strategy, however, does not generalize well to more complex 3D scenes where ray paths do not align with predefined cells, leading to rendering artifacts or training challenges.
This paper proposed a novel strategy to overcome these limitations, by introducing a more flexible KD-tree partitioning method and optimized multi-GPU training architecture to enhance the applicability of GS models across varied 3D scenes.

**Strengths:**

+ Significance: this paper is well motivated and tackles a critical gap in existing GS-based methods by addressing the need for higher resolution and the capacity to handle larger datasets and more complex scenes. By moving towards scalable, high-resolution 3D reconstruction, this work could influence a range of fields such as virtual reality, autonomous driving, and simulation.
+ Quality: the distributed training of 3DGS is well supported by theory and proved efficacy in the experiments. Illustrations are clear and straightforward, with abundant experiments on diverse scenes. Supplemantary material is informative and solid.

**Weaknesses:**

- Missing related work: the proposed partitioning and ray segment rendering reminds me of an existing work NeRF-XL. Perhaps these are concurrent works, but authors should at least discuss it in related work.
- Practical usage of KD-tree: it seems that KD-tree partition makes sense when the scene or scene points are evenly distribuuted. Will it still maintain a balanced workload between workers when the scene is not evenly distributed.
- MVS points for initialization: I suppose that authors need to work from a quite dense and envenly distributed point initialization so they need to do MVS first. This is not necessary a weakness but maybe constraining its application e.g. when views are sparse or inconsistent to get an ideal MVS result, which is quite common in large-scale scene capturing.

**Questions:**

This work is developed upon 3DGS, can it be compatible with derivatives of 3DGS, e.g. 2DGS or scaffoldgs?

---

> ### Author Response · Authors · 2024-11-27
>
> **Q1: Missing related work（e.g. Nerf-XL）**
>
> Thank you for your suggestion. We will include the mentioned works in the Related Work section and discuss them accordingly.
> Nerf-XL primarily addresses distributed training for NeRF and adopts a partial rendering strategy. However, it is important to emphasize that applying partial rendering in 3DGS is non-trivial and significantly more complex than in NeRF-based methods.
> - First, the rendering primitive in 3DGS, splats, has volume, while the primitive in NeRF, sampling points, has no volume. Splats at the boundaries of sub-models pose challenges for sorting in alpha blending and for the partitioning of model parameters for parallelism. A detailed discussion of this is included in the "Novelty and Value" section of the "Official Comment" visible to all reviewers.
> - Second, in NeRF-based methods, a uniform subdivision of the scene ensures load balancing. In 3DGS, however, splat distributions are often uneven, necessitating more sophisticated strategies to balance the computational workload among sub-models. We employ a KD-tree for this purpose and have demonstrated its effectiveness in our experiments.
>
> **Q2: Practical usage of KD-tree**
>
> In fact, we partition the KD-tree based on the number of splats, which ensures that the number of splats in each block is approximately equal. As a result, it can adapt to situations where the point distribution is uneven. Table 2 demonstrates that this strategy performs much better than a uniform partitioning approach (based on spatial volume).
>
> **Q3: MVS points for initialization**
>
> Thanks for raising this point. In the Mip-NeRF360 Garden dataset,  if the original densification strategy in 3DGS is used, the model still converges with no significant impact on PSNR (27.33 vs. 27.41). However, the original densification strategy in our experiments can not achieve the desired large number of primitives, a limitation we have thoroughly discussed in Section 4.2 of the paper.  In other words, the precision provided by MVS is not strictly necessary; a faster but less precise MVS solution (e.g., DUSt3R) could be used to provide a dense initialization, and then the adaptive capabilities of 3DGS could drive convergence. In the MatrixCity-Aerial dataset, we randomly selected 10% of the views for MVS initialization, keeping all other conditions the same. The model converged normally, and the PSNR showed only a small decrease compared to using MVS initialization with all views (27.70 vs. 27.65).
>
> **Q4: This work is developed upon 3DGS, can it be compatible with derivatives of 3DGS, e.g., 2DGS or ScaffoldGS?**
>
> Yes. Thanks for this great suggestion. The directives of 3DGS, e.g. 2DGS and ScaffoldGS, use volumetric primitives and alpha blending for rendering The proposed strategy can apply to these types of models and enable distributed training of them.

---

### Official Review · Reviewer_MeKu · 2024-11-01

**Soundness:** 3
**Presentation:** 3
**Contribution:** 3
**Rating:** 5
**Confidence:** 4

**Summary:**

This paper introduces a distributed training framework, RetinaGS, for training large-scale 3D Gaussian Splatting (3DGS) models to achieve high-definition 3D scene reconstruction. To overcome the limitations of single-GPU training, RetinaGS employs a precise distributed rendering equation, uses KD-tree partitioning for load balancing, and initializes Gaussian splats through multi-view stereo, enabling parallel training of 3DGS across multiple GPUs. Experiments were conducted on datasets including Scannet++, MipNeRF-360, Mega-NeRF, and MatrixCity, demonstrating superior rendering quality and efficiency of RetinaGS on large-scale datasets. Notably, RetinaGS achieved the training of over a billion splats on the MatrixCity dataset, reaching unprecedented visual quality.

**Strengths:**

1. Clear Writing: The paper is well-organized, with logically structured sections and clear language. It effectively presents the problem background, research motivation, methods, and experimental results, making it easy for readers to understand the design principles and implementation of RetinaGS.
2. Engineering Contributions of Distributed Training: The proposed distributed training framework is technically sound and addresses high-performance computing needs. The framework also makes notable engineering contributions, including the use of KD-tree for load balancing across GPUs and multi-view stereo (MVS) for initializing Gaussian splats, which improve the efficiency and stability of distributed training.
3. Broad Experimentation: The method was tested on multiple datasets, including Scannet++, MipNeRF-360, Mega-NeRF, and MatrixCity, showcasing its applicability and performance across varying scene scales. The MatrixCity dataset especially highlights the scalability and rendering quality of the method.

**Weaknesses:**

1. Limited Scale of Experimented Scenes: Although the method was tested on four datasets, Scannet++ and MipNeRF-360 have relatively small scene scales, limiting the demonstration of its potential for large-scale scenes. Moreover, while MatrixCity is a large-scale dataset, it is synthetic and lacks validation on real-world scenes. Testing on more large-scale datasets, especially with real-world data, would better illustrate the practical applicability of RetinaGS.
2. Lack of Comparison with Other Methods: The paper does not include comparisons with methods like VastGaussian or Hierarchical 3D Gaussians, which are also significant contributions in distributed 3D reconstruction. Comparing RetinaGS with these methods would provide a more comprehensive view of its advantages and limitations.
3. Lack of Quantitative Results in Exploration Study: In the first two sections of the EXPLORATION STUDY, the lack of quantitative results weakens support for the effectiveness of the method. Adding quantitative analysis would provide a clearer demonstration of the impact of different settings on model performance.
4. Limited Novelty: The core idea of this paper is relatively straightforward, with the primary contribution being in the engineering implementation rather than in methodological innovation. Consequently, RetinaGS demonstrates moderate novelty in terms of the underlying approach.

**Questions:**

Insufficient Scenes in Mega-NeRF Experiments: The paper uses only three of the six scenes from the Mega-NeRF dataset for experiments, without explaining the rationale for this choice. This may affect the comprehensiveness of the experimental results. It would be beneficial to include results from all scenes or provide a clear explanation for the selected subset.

Some minor issues:
1. The word “distirbuted” should be corrected to “distributed” in Line 97.
2. In Table 4 on page 9, “Bacth Size” should be corrected to “Batch Size.”

---

> ### Author Response · Authors · 2024-11-27
>
> **Q1: Limited Scale of Experimented Scenes**
>
> MatrixCity-ALL is currently the largest known dataset, containing 140k images, whereas other public datasets typically only have a few thousand images, making the scale difference around 100 times. At present, few works have tackled this dataset directly. Although MatrixCity is synthetic, we still use COLMAP to obtain initial camera poses, which has similar pose estimation errors that might occur in real-world data. Furthermore, collecting large-scale real-world scenes is very costly. We evaluate parallelization strategies for large-scale 3DGS models using the synthetic MatrixCity in this work but also look forward to new contributions of large-scale real-world datasets in the community.
>
> **Q2: Lack of Comparison with Other Methods (eg. VastGaussian or Hierarchical 3D Gaussians)**
>
> We compare RetinaGS with VastGaussian and HierarchicalGS on the Mega-NeRF rubble. RetinaGS achieves a PSNR of 27.42, outperforming VastGaussian, which reports a PSNR of 26.92, and HierarchicalGS, which reports a PSNR of 24.64.
>
> **Q3: Lack of Quantitative Results in Exploration Study**
>
> This section discusses three parts in total. The quantitative results for "Initialization and Densification" are presented in Fig. 6, where we compare our strategy with the original 3DGS strategy using both curve and scatter plots. The quantitative metrics and visualization results for "Validity of Distributed Rendering" are shown in Fig. 7. Finally, the quantitative results for "KD-Tree Partition" are provided in Table 2.
>
> **Q4: Limited Novelty**
>
> We highlight the discussion on novelty and contribution in "Novelty and Value" section of the "Official Comment," which is visible to all reviewers. Please refer to the relevant section for more details.
>
> **Q5: Insufficient Scenes in Mega-NeRF Experiments**
>
> This is a common practice in the community. CityGaussian also selected these same three scenes in their main paper and provided evaluation results for Mega-NeRF, Switch-NeRF, GP-NeRF, 3DGS, and other methods on these scenes. By using the same metrics, we ensure a fair comparison with other approaches.

---

### Official Review · Reviewer_oADR · 2024-11-02

**Soundness:** 3
**Presentation:** 3
**Contribution:** 2
**Rating:** 5
**Confidence:** 5

**Summary:**

This paper, named RetinaGS, aims to train 3DGS in a distributed way. It first divides the scene space into a set of convex subspaces, each subspace contains a subset of 3DGS and can thus be distributed trained. For each subspace, the proposed method calculates a partial color and partial opacity, and the final color is obtained by a weighted sum of all subspaces. This paper also did exhaustive experiments to show the effectiveness of their method on the MipNeRF360 dataset, Mega-NeRF dataset, and the MatrixCity dataset. The proposed method can achieve comparable rendering quality to existing NeRF/3DGS methods (GP-NeRF, 3DGS, and CityGaussian), while with many more model parameters.

**Strengths:**

(1) This paper did very exhaustive experiments on publicly available large-scale datasets, such as Mega-NeRF and MatrixCity.

(2) This paper can train 3DGS up to billion-scale 3D Gaussian primitives, which is very impressive.

(3) This paper is a good engineering work, though it is not the only work that can scale up the training of 3D GS to billion-scale data.

**Weaknesses:**

(1) This method lacks discussion of more existing distributed methods, such as Hierarchical-GS (Kerbl, *el,al*, SIGRRAPH 2024), DOGS (Chen and Lee, NeurIPS 2024), and City-on-Web (Song *et,al*, ECCV 2024).

(2) To me, this method is quite similar to 1) **DOGS**, which also adopts a central manager to manage sub-models, and 2) **City-on-Web**, which also adopts a similar partial color and partial opacity rendering strategy (while City-on-Web uses volume rendering instead of point rendering) to compute the final color. Therefore, discussions and comparisons to DOGS and City-on-Web are necessary while missing in the current version.

(3) The proposed method requires a gradient computation step and a gradient synchronization step on the central manager, which can consume more time than other distributed methods (VastGaussian, DOGS). The time for these steps should be clarified in the paper.

(4) This method requires running MVS on all training data. However, running MVS on large-scale datasets is time-consuming, and can even require much more time than training 3DGS on the same dataset.

(5) This paper cares about the distributed training of 3DGS, and how the method can scale up the training of 3DGS to a billion scale. The improvements in visual quality are only based on the improved number of 3DGS. e.g. in Table 1, the PSNR is 27.70 with 217.30M 3D Gaussians, which has only marginal improvement compared to CityGaussian with 23.7M 3D Gaussians.

(6) I appreciate the authors' effort in running exhaustive experiments on many existing large-scale datasets. However, the paper is more like a system work, and the novelty of the paper is limited, especially since the partial color and partial opacity strategy are already proposed in City-on-Web and there is lack of discussion of this paper with related works.

**Questions:**

(1) In the experiments, RetinaGS uses dense point clouds for initialization. Do the other methods use the same dense point clouds for a fair comparison? If not, the author should provide their results with sparse/dense point clouds.

(2) The author provides training time for their method in Table 2 and Table 3. While the training time of other methods is missing. I wonder would RetinaGS would be slower/faster than CityGaussian/VastGaussian/DOGS?

---

> ### Author Response · Authors · 2024-11-27
>
> **Q1: Lacks discussion of more existing distributed methods…**
>
> Thank you for pointing this out. We will include the mentioned works in the Related Work section and provide a detailed discussion. Specifically, you highlighted the connections between our work and DOGS and City-on-Web. Below, we discuss these works and their differences from ours:
>
> - **City-on-Web** primarily addresses distributed training for NeRF and adopts a partial rendering strategy. However, it is important to emphasize that applying partial rendering in 3DGS is non-trivial and significantly more complex than in NeRF-based methods.
> First, the rendering primitive in 3DGS, splats, has volume, while the primitive in NeRF, sampling points, has no volume. Splats at the boundaries of sub-models pose challenges for sorting in alpha blending and for the partitioning of model parameters for parallelism. A detailed discussion of this is included in the "Novelty and Value" section of the "Official Comment" visible to all reviewers.
> Second, NeRF-based methods use uniform scene subdivision for load balancing, but 3DGS's uneven splat distribution requires more advanced strategies. We address this with a KD-tree, proving its effectiveness in our experiments.
>
> - **DOGS** and VastGaussian/CityGaussian use a similar strategy of dividing training data and splats into predefined blocks, independently training 3DGS sub-models for each block. While this differs from our method (see Fig. 2), it approximates native 3DGS and may cause boundary artifacts. DOGS mitigates this with overlapping regions and boundary synchronization but assumes each ray/view belongs to one sub-model—a valid approximation for bird's-eye datasets. For general datasets, where rays intersect multiple sub-models, significant non-adjacent information must be cached, increasing parameter demands and reducing scalability as DOGS' model parallelism devolves into data parallelism.
>
>
> **Q2: The proposed method requires a gradient computation step and a gradient synchronization step...**
>
> The gradients that need to be computed can be divided into loss gradients and sub-model gradients. As shown in Equation (5), the central manager only computes and synchronizes the loss gradients with respect to the partial color and opacity, which a 2D map with the same resolution as the image can represent. This means that the required communication bandwidth and computation are negligible. According to our statistics, the time overhead for this step accounts for less than 1% of the total time. Additionally, a small portion of the splats is shared between workers in our implementation, requiring gradient synchronization concerning these primitives. This synchronization takes less than 5% of the total step time in our experiments. So, overall, we believe the overhead introduced by distributed training is minimal in our method.
>
> **Q3: Running MVS on large-scale datasets is time-consuming…**
>
> We want to emphasize that the primary purpose of using MVS in this paper is to obtain a sufficient number of Gaussian splats. The original densification strategy in 3DGS can be also used when the model still converges with no significant impact on PSNR. However, the original densification strategy in our experiments can not achieve the desired large number of primitives, a limitation that we have thoroughly discussed in Section 4.2 of the paper. A densification strategy that enables GS points to scale to large sizes is another important topic, but beyond the scope of this paper.
>
> **Q4: … has only marginal improvement compared to CityGaussian ...**
>
> We highlight the discussion in "Improvement is not Marginal" section of the "Official Comment," which is visible to all reviewers. Please refer to the relevant section for more details.
>
> **Q5: ...the author should provide their results with sparse/dense point clouds.**
>
> As shown in Fig. 4, when the number of splats is the same, the PSNR of 3DGS and RetinaGS, which only differ in initialization and densification, are almost identical, indicating that RetinaGS does not rely on MVS's potential higher initialization accuracy to achieve the same accuracy as 3DGS. We use MVS solely to obtain a sufficiently dense set of splats, as the native densification strategy cannot easily increase the number of GS points as we desire.
>
> **Q6: I wonder would RetinaGS would be slower/faster than CityGaussian/VastGaussian/DOGS?**
>
> Our approach is faster. As shown in the discussion above, communication overhead accounts for only about 5% of the total step time. The true bottleneck in distributed 3DGS training lies in the load balancing strategy. Other methods overlook load balancing, resulting in uneven splat distribution and significant computational disparities across sub-models, which cause idle waiting times. Table 2 demonstrates that with our KD-Tree load balancing strategy, we reduced training time from 32.4s to 18.9s when using 4-node parallelism, achieving a speedup of 71.4%.

---

> ### Comment · Reviewer_oADR · 2024-11-28
> **Thanks for the rebuttal**
>
> Thanks the authors for their rebuttal. While most of my concerns are resolved, I prefer to keep my score currently and will consider to raise my score with further discussion.
>
> As the authors claimed they use MVS point clouds for initialization to achieve a desired number of Gaussian primitives on the MatrixCity street view scenes, I just realized that most of the experiments of Hierarchical-GS (Kerbl, el,al, SIGRRAPH 2024) are done on street view scenes. Moreover, Hierarchical-GS also adopts a tree structure to enable the distributed chunk training of 3DGS, which is similar to the KD-tree proposed in this work. Therefore, I would like the authors:
> 1) to further **highlight the similarities and differences of the method with Hierarchical-GS**.
> 2) **compare with Hierarchical-GS on the street view scenes used by Hierarchical-GS under the same initialization settings** (I believe Hierarchical-GS uses sparse points from COLMAP for initialization on the street view scenes).
>
> I would raise my score if the proposed method can achieve comparable results to Hierarchical-GS.

---

> > ### Author Response · Authors · 2024-12-04
> >
> > We greatly appreciate your response and valuable suggestions. We will incorporate the relevant discussions and experimental results into the main text.
> >
> > **Q1：to further highlight the similarities and differences of the method to Hierarchical-GS.**
> >
> > HierarchicalGS introduces the concept of Level of Detail (LOD) to 3DGS, significantly reducing the number of splats actually involved in rendering for each view and thereby improving rendering speed. There are two key distinctions to highlight here:
> > 1. Focus on scaling directions: HierarchicalGS focuses on “scaling in” by reducing the computational load of a single sub-model. In contrast, RetinaGS focuses on “scaling out” by enabling parallel training of more sub-models, thereby expanding the system's scale. In large-scale models, such as LLMs, scale-in and scale-out strategies are not mutually exclusive. In fact, they should often be employed together to maximize system scalability.
> > 2. Memory and computational costs: In HierarchicalGS, all splats are still maintained as leaf nodes of the hierarchical tree, and additional parent nodes must also be maintained. As a result, GPU memory requirements do not decrease but rather increase ( ≈68% larger than non-hierarchical 3DGS), potentially requiring more computational nodes to accommodate the same model scale. This highlights the necessity of introducing a scale-out solution like RetinaGS.
> >
> > **Q2：compared with Hierarchical-GS on the street view scenes used by Hierarchical-GS under the same initialization settings (I believe Hierarchical-GS uses sparse points from COLMAP for initialization on the street view scenes).**
> >
> > Most of the experiments in HierarchicalGS were conducted on a custom dataset consisting of four street-view scenes created by the authors. Currently, only the smallest scene, named SmallCity, has been released, which contains only a few thousand images. Additionally, they reported experimental results on publicly available datasets of a similar scale (Building and Rubble scenes provided by Mega-NeRF). We chose these datasets for our comparative experiments. The experimental results are as follows:
> >
> > | Method                                   | Building | Rubble |
> > |------------------------------------------|----------|--------|
> > | RetinaGS + MVS                           | **22.14**    | **25.77**  |
> > | RetinaGS + sparse initialization         | **21.83**    | **24.85**  |
> > | HierarchicalGS + sparse initialization + depth prior   | 21.52    | 24.64  |
> >
> > Even with sparse initialization, RetinaGS achieves better PSNR. It is worth noting that HierarchicalGS additionally uses depth information as supervision. According to the results reported by HierarchicalGS, rendering with only leaf nodes consistently produces better visual quality compared to using the hierarchy. When the number of leaf nodes is the same, the hierarchy only provides faster rendering and training speeds but at the cost of reduced rendering quality and increased memory usage.

---

### Author Response · Authors · 2024-11-27

We sincerely thank all four reviewers for their thorough reading, valuable suggestions, and recognition of the completeness of our experimental results and implementation. We will correct the typographical errors, fill in missing details, and clarify ambiguous statements mentioned by the reviewers. Now, let us first highlight the novelty and value of this work.

**Novelty and Value:**

Both NeRF and 3DGS use alpha blending to aggregate the influence of a large number of primitives. It is known that the computation of alpha blending can be equivalently decomposed into multiple partial renderings, which are then fused into the final result. However, the realization of this separability, which is crucial for model-parallel distributed training, is different for NeRF and 3DGS.

For NeRF, as the rendering primitives in NeRF are sampling points, which are volumeless and static, the separation point for each ray is the intersection point of the ray and the subspace boundary. Therefore, dividing the space into non-overlapping subspaces and assigning different MLP models to handle the sampling points based on their coordinates achieves model parallelism.

For 3DGS, however, this is not as straightforward. The rendering primitives in 3DGS are Gaussian splats, which have volume and are dynamic. When splats span multiple subspaces, ensuring that these splats are rendered correctly in each subspace to maintain the correct alpha blending order in the final result becomes a significant challenge that is easy to overlook. We validate this in Fig. 6 of the paper, showing that naive handling leads to severe rendering errors at the boundaries in typical scenes (excluding bird-eye view datasets).

Our approach realized the separability and achieved distribution agnostic model parallelism for 3DGS. We expanded the subspace boundaries to allow for partial sharing of splats. With the indicator function defined in Eq. 6, we can efficiently determine whether a splat should participate in rendering the current subspace for each ray with negligible computational cost and without per-primitive communication in the forward and backward computation.

The rendering equations modified by the indicator function in Eq. 3 and Eq. 4 ensure that rendering within a subspace remains computationally efficient. Additionally, Appendix Section 2 proves that Eq. 3 to Eq. 5 are strictly equivalent to the native 3DGS rendering equations. We not only preserve the separability of alpha blending calculations but also ensure the decomposability of 3DGS model parameters, thereby achieving efficient model parallelism distributed training for 3DGS.

We summarize the novelty and value of RetinaGS below and hope that the reviewers will recognize our efforts and the significance of this work. We will also include a detailed discussion of these points in the main text:
1. Previous parallel 3DGS works have only validated their effectiveness on bird-eye view datasets. However, the strict equivalence of decomposability cannot be overlooked in general cases, as it would otherwise lead to significant visual errors at the boundaries. The proposed method does not assume any type of view-points or distribution of Gaussian primitives.
2. Load balancing between sub-models is also non-trivial for 3DGS. Unlike Nerf, the distribution of splats in 3DGS is uneven, which can lead to GPUs waiting on each other. We introduce KD-Tree to address this issue and achieve proper load balancing in training and rendering.
3. The strict equivalence ensures that the resulting 3DGS model is equivalent to a single large 3DGS model. The model can be consolidated onto one GPU if the memory allows or repartitioned to an arbitrary number of devices without any quality loss. In contrast, the compared 3DGS or NeRF based distributed training methods result in multiple separate sub-models that cannot be consolidated or repartitioned without significant change in implementation.
4. Thanks to the scalability of the proposed method, we, for the first time, can explore the training of 3DGS models on the level of more than one billion primitives. This resulted to unprecedented quality improvement in large-scale scenes with arbitrary viewpoints.

**Improvement is not Marginal in General Large Dataset:**

Due to earlier GS-based distributed methods, such as CityGaussian and VastGaussian, which were tested only on small-scale aerial view datasets (e.g., MatrixCity-Aerial with **6k** images), and their native implementations currently being challenging to scale to larger, more general datasets, the only SOTA work we found that reports experiments on a complete street-view dataset is DOGS, which evaluated MatrixCity-Street with **140k** images. DOGS reports a PSNR of **21.61** on MatrixCity-Street with 2.37M splats. In contrast, we successfully scaled the number of splats to the billion level, achieving a PSNR of **25.50** (Note that PSNR is a logarithmic metric).

---

### Meta-Review · Area_Chair_9c9R · 2024-12-17

**Metareview:**

This paper receives ratings of 5,5,6,5. The AC follows the recommendations of the reviewers to reject the paper. The main concerns of this paper are: 1) The lack of discussion on more existing distributed methods. 2) The proposed method is very similar to existing works such as DOGS and City-on-web. 3) Performance and efficiency might not be as good as other existing methods. 4) Limited scale of experimented scenes and lack of ablations. Although the authors tried to provide their rebuttals to address the issues, the reviewers remained concern over the proposed method and experiments.

**Additional Comments On Reviewer Discussion:**

Although the authors tried to provide their rebuttals to address the issues, the reviewers remained concern over the proposed method and experiments.

---

### Decision · Program_Chairs · 2025-01-22

Reject